# Potential of Long Non-Coding RNAs in Age-Related Macular Degeneration

**DOI:** 10.3390/ijms22179178

**Published:** 2021-08-25

**Authors:** Janusz Blasiak, Juha M. T. Hyttinen, Joanna Szczepanska, Elzbieta Pawlowska, Kai Kaarniranta

**Affiliations:** 1Department of Molecular Genetics, Faculty of Biology and Environmental Protection, University of Lodz, Pomorska 141/143, 90-236 Lodz, Poland; 2Department of Ophthalmology, University of Eastern Finland, 70210 Kuopio, Finland; Juha.Hyttinen@uef.fi; 3Department of Pediatric Dentistry, Medical University of Lodz, 92-216 Lodz, Poland; joanna.szczepanska@umed.lodz.pl; 4Department of Orthodontics, Medical University of Lodz, 92-217 Lodz, Poland; elzbieta.pawlowska@umed.lodz.pl; 5Department of Ophthalmology, Kuopio University Hospital, 70210 Kuopio, Finland

**Keywords:** age-related macular degeneration, AMD, long non-coding RNA, epigenetic regulation, oxidative stress-induced dedifferentiation, micro RNA

## Abstract

Age-related macular degeneration (AMD) is the leading cause of visual impairment in the aging population with poorly known pathogenesis and lack of effective treatment. Age and family history are the strongest AMD risk factors, and several loci were identified to contribute to AMD. Recently, also the epigenetic profile was associated with AMD, and some long non-coding RNAs (lncRNAs) were shown to involve in AMD pathogenesis. The Vax2os1/2 (ventral anterior homeobox 2 opposite strand isoform 1) lncRNAs may modulate the balance between pro- and anti-angiogenic factors in the eye contributing to wet AMD. The stress-induced dedifferentiation of retinal pigment epithelium cells can be inhibited by the ZNF503-AS1 (zinc finger protein 503 antisense RNA 2) and LINC00167 lncRNAs. Overexpression of the PWRN2 (Prader-Willi region non-protein-coding RNA 2) lncRNA aggravated RPE cells apoptosis and mitochondrial impairment induced by oxidative stress. Several other lncRNAs were reported to exert protective or detrimental effects in AMD. However, many studies are limited to an association between lncRNA and AMD in patients or model systems with bioinformatics. Therefore, further works on lncRNAs in AMD are rational, and they should be enriched with mechanistic and clinical studies to validate conclusions obtained in high-throughput in vitro research.

## 1. Introduction

Does everything lie in genes? In a way, yes, as the genetic constitution of an organism determined by the sequence of its genome is a blueprint for an individual, but it represents the only potential for its development in normal conditions. The final phenotype is determined by the pattern of gene expression, which is underlined by both the genome sequence and its epigenetic modifications, as well as the environment/lifestyle that may also influence phenotype by non-genetic mechanisms. Changes in the epigenetic pattern are associated with many pathological states, and they are an essential step in cancer transformation. The epigenetic pattern is established by DNA methylation/demethylation, histone modifications, and the action of regulatory non-coding RNAs, which can be broadly divided into short and long non-coding RNAs (sncRNAs and lncRNAs, respectively).

Age-related macular degeneration (AMD) is a common cause of vision loss in the elderly. It is a complex, multifactorial eye disease with an interplay between genetic and environmental/lifestyle factors in its pathogenesis. AMD epigenetics is relatively recently subjected to AMD research but exploding during the last 10 years. The first paper related to AMD epigenetics from Salminen’s and Kaarniranta’s lab appeared in 2007, but in 2011 Hjelmeland still called epigenetics in AMD as a “dark matter” [1,2]. Today (22 July 2021), searching PubMed, even with a simple syntax “epigenetic AMD or epigenetic retina degeneration”, returns 181 results with a significant portion published in the last three years. There are several recent excellent reviews on epigenetics in AMD, but lncRNAs are likely the least frequently addressed subject due to a low number of experimental works on lncRNAs in AMD. In this review, we update information on the involvement of lncRNAs in AMD pathogenesis and present the potential of lncRNAs in AMD diagnosis and therapy. In some cases, the results of works on other retinal diseases, especially diabetic retinopathy (DR), are presented as they contain information on retinal changes that can also contribute to AMD pathogenesis, e.g., retinal neovascularization. Many studies are allocated as “diabetic retinopathy” on the basis of oxidative stress induced by glucose at high concentrations, but this stress can also contribute to AMD pathogenesis [3].

## 2. Age-Related Macular Degeneration (AMD)—A Complex Eye Disease with the Involvement of Genetic/Epigenetic Factors in Its Pathogenesis

Age-related macular degeneration, a neurodegenerative disease of the eye, is the commonest reason for legal blindness and vision loss in the elderly. Its worldwide estimated prevalence in 2040 is about 300 million [4]. It is associated with physical and mental problems for affected individuals as well as it is a serious burden for societies. AMD affects the macula, a small functional structure in the central retina responsible for fine and color vision. The disease affects vision through progressive damage and loss of photoreceptors and the underlying retinal pigment epithelium (RPE). The RPE plays an important role in the maintenance of the balance in the retinal microenvironment through phagocytosis of the used photoreceptor outer segments (POSs), involvement in the visual cycle, and the blood-retina barrier as well as secreting growth factors needed for endothelial cells (reviewed in the work of [5]). Due to this and other effects, RPE cells are a main player in AMD pathogenesis.

Clinically, AMD can be divided into two forms: dry, which may progress to geographic atrophy (GA), and wet, featured by choroidal neovascularization (CNV) (Figure 1). Fibrotic signs are typical in the advanced wet AMD. Although the exact mechanism of AMD pathogenesis is not fully known, the formation of drusen associated with pigment mottling is evoked by impaired RPE functions [6]. Then, the disease may progress to GA with drusen resorption and hypopigmentation caused by RPE loss. In wet AMD, the reduction in blood supply induced by stenosis of large blood vessels may lead to the loss of choroidal vasculature and the creation of a proinflammatory environment [7]. In wet AMD, RPE is a key player to produce angiogenic compounds, including vascular endothelial growth factor (VEGF), which is targeted by specific antibodies or fusion proteins in the therapy of this AMD form.

Age is, per definition, the most serious factor of AMD pathogenesis, and several other reported or putative factors may play a role. These include modifiable factors, such as tobacco smoking, improper diet, high body-mass index, increased blood lipid, and cholesterol levels, female sex, exposure to the sunlight, especially its blue component, and others (reviewed in the work of [8]). Almost all these factors may contribute to the production of reactive oxygen species (ROS) and oxidative stress, which is frequently presented as the main factor of AMD pathogenesis.

Mitochondrial dysfunction in the RPE has been shown to play a role in AMD pathophysiology (reviewed in the work of [9,10,11]). It was shown that postnatal ablation of RPE mitochondrial oxidative phosphorylation in mice triggered epithelium dedifferentiation, characterized by cellular hypertrophy, reduction in some RPE proteins, and photoreceptors degeneration [12]. RPE cells dedifferentiation and hypertrophy were induced by stimulation of the AKT (AKT serine/threonine kinase)/mTOR (mechanistic target of rapamycin). Administration of an oxidant to wild-type mice resulted in RPE cells dedifferentiation and mTOR activation. In contrast to the normal retina, dedifferentiated RPE cells could proliferate but were not able to complete cytokinesis. Such an abnormal proliferation was also observed in human RPE cells adjacent to drusen [13]. In addition, transdifferentiation of RPE cells was associated with CNV in samples excised from AMD-related choroidal neovascular membranes [14]. Rapamycin, an mTOR inhibitor, reduced dedifferentiation and preserved photoreceptor functions. Although significant changes in some autophagy markers in the modified animals were not observed in that study, that important work suggests that dedifferentiation of the RPE cells might be induced by oxidative stress and the mTOR signaling, essential in autophagy, can contribute to AMD pathogenesis and can be considered as a potential therapeutic target in this disease. Although mice do not have maculae, the processes of atrophy and degeneration occurring in their retina are similar, if not the same, to the processes occurring in the human retina.

Positive family history is, apart from aging, the strongest AMD risk factor. Family segregation and twin studies performed by Seddon JM et al. suggest that genetic components may play a role in AMD and age-related maculopathy, and up to 71% of AMD cases may be underlined by genetic factors [15]. Early association and genetic linkage studies identified the complement factor H (CFH) gene and the age-related maculopathy susceptibility 2/HtrA serine peptidase (ARMS2/HTRA1) genes to be associated with AMD [16]. Subsequent studies with high-throughput techniques, first of all, GWAS (genome-wide association studies), confirmed these conclusions and revealed a genome-wide set of other common variants of complement-related genes associated with AMD occurrence, including C2/CFB, C3, C7, CFI, and SERPING (reviewed in the work of [17]). However, the results of GWAS published in 2013 by the AMD Gene Consortium identified 19 loci showing enrichment in genes involved in the regulation of complement activity, collagen synthesis, lipid metabolism/cholesterol transport, receptor-mediated endocytosis, endodermal cell differentiation, angiogenesis, and extracellular matrix organization were associated with AMD, but only up to 15–65% of all AMD cases might be attributed to those variants [18]. The discrepancy between that and earlier studies was attempted to be explained by the interplay with environment, rare variants, gene-gene interaction, and epigenetics.

If the genetic profile is important in AMD, the same concerns its epigenetic counterpart, as any phenotype is not determined by gene sequence per se, but the gene expression regulated by epigenetic marks and events. However, the epigenetic profile may play a role in AMD pathogenesis as it may influence the expression of genes whose sequence is not changed, but their alternate products are important in AMD. As the DNA sequence of an individual is difficult to modify, the epigenetic profile can be alternated by many drugs and environmental factors and even with the lifestyle, especially diet [19]. This creates an opportunity to consider modifications of the epigenetic profile not only in experimental research on AMD pathogenesis but also in AMD prevention and therapy [20,21].

Three main epigenetic regulatory elements are DNA methylation, post-translational histone modification, and non-coding RNAs. Micro RNAs (miRNAs) are the most intensively investigated sncRNAs also in AMD (reviewed in the work of [22]). However, lncRNAs can act as a sponge (sink) for many miRNAs and, in this respect, may be superior to miRNAs in gene regulation. Despite this, studies on lncRNAs are much less intense than on miRNAs, also in AMD. This is likely due to poorer knowledge and higher experimental requirements of lncRNAs as compared with miRNAs.

## 3. Long Non-Coding RNAs

Only a small proportion of the human genome contains protein-coding genes, but almost the entire genome is transcribed [23]. Apart from pre-mRNAs, the genome transcriptome constitutes various lncRNAs, which can be defined as RNA species longer than 200 nt that are not translated as mostly they have no open reading frame and translation initiation signals. However, some transcripts annotated as lncRNAs code for small peptides [24,25,26]. Most lncRNAs are in the nucleus, but about 15% can be found in the cytoplasm [27].

Long non-coding RNAs may be classified in many ways in dependence on their length, functions, and biogenesis, the location of their genes, relationship with protein-coding genes, subcellular localization, and additional characteristics (reviewed in the work of [7]). According to their origin, lncRNAs can be divided into large intervening/intergenic non-coding RNAs (lincRNAs), transcribed by Pol II from intergenic regions, are presumably capped, spliced and polyadenylated, antisense transcripts of protein-coding genes (natural antisense transcripts (NATs)), snoRNA (small nucleolar RNA)-ended lncRNAs (sno-lncRNAs) are derived from excised introns, some lncRNAs are processed by RNase P and stabilized by the U-A-U triple helix, 5’ snoRNA-ended and 3’-polyadenylated lncRNAs (SPAs), circular intronic RNAs (ciRNAs) derived from excised introns and depend on consensus RNA sequences to avoid debranching of the lariat introns, circular RNA (circRNAs) are produced by back-splicing circularization of exons of pre-mRNAs (reviewed in the work of [28]).

Long non-coding RNAs may be classified in many other ways, and frequently, they are categorized as intergenic lncRNAs, intronic lncRNAs, bidirectional lncRNAs, overlapping sense lncRNAs, and antisense lncRNAs, lncRNAs hosted by a miRNA gene or cluster of miRNA genes (Figure 2) [29]. Their genes are located throughout the genome, but they are frequently found in the regions of other genes located in their exons, introns, and regulatory sequences, including promoters and enhancers. They may adopt open (linear) and closed (circular) conformations. In animals, most of the lncRNAs are produced by RNA polymerase II, but some are transcribed by RNA polymerase III or I [30,31].

In general, the fundamental function of lncRNAs is to regulate gene expression that can be performed directly or indirectly in *cis* or *trans* [32]. Sometimes it is not easy to distinguish the functions performed by a specific lncRNA from the function playing by the gene from which it is transcribed. The most established example of *cis*-acting lncRNA is XIST (X-inactive specific transcript) that is responsible for inactivating one X chromosome during female mammal development [33,34,35]. However, despite long-lasting and many intense works on the mechanism of X-inactivating action of XIST, not all aspects of its influences are known, which reflects a general multi-mode action of lncRNAs [32]. Sometimes, the transcription of the lncRNA locus can be essential for the regulation of gene(s) located at or in the close vicinity of that locus and the antisense lncRNA, Airn (antisense Igfr2 (insulin-like growth factor 2 receptor) RNA non-coding) overlapping the *Igfr2* gene, is essential in the silencing its paternal allele, but it does not require any DNA elements within its locus to exert its imprinting action [36]. Transcription of a lncRNA locus can also affect transcription of closely located genes by interacting with RNA polymerase or/and affecting transcription factors binding and modification of corresponding chromatin domains. Some mutations in the Blustr (linc1319) lncRNA resulting in the deregulation of its promoter, polyA signal, and the 5′ splice site led to a decreased expression of the *Sfmbt2* (Scm-like with four mbt domains 2) gene [37]. These examples illustrate situations when the *cis*-regulatory activity of a lncRNA locus is due to *cis*-acting corresponding lncRNA, but in some cases, such activity can be attributed to DNA elements within the locus. The lincRNA-p21 was identified as a p53-dependent lncRNA produced from an intergenic locus located near the *CDKN1A* (cyclin-dependent kinase inhibitor 1A) gene [38]. However, further studies revealed that this lncRNA regulated *CDKN1A* in *cis* [39]. Later research showed that the lincRNA-p21 locus contained *cis*-regulatory DNA elements that modulated the expression of neighboring genes [40]. Some lncRNAs leave the site of their transcription and perform their functions in *trans* [41]. Three major subgroups of such lncRNAs can be distinguished: regulating gene expression and chromatin organization away from their transcription site, influencing the nuclear structure, and interacting with proteins and other RNA species. One of the first lncRNAs discovered as regulating gene expression in trans was HOTAIR (HOX antisense intergenic RNA), whose involvement in breast carcinogenesis is suggested (reviewed in the work of [42]). After HOTAIR, many other lncRNAs were discovered to regulate gene expression in *trans,* and this effect is not limited to transcripts of RNA polymerase II as SLERT (snoRNA-ended lncRNA enhances pre-ribosomal RNA transcription) was demonstrated to affect transcription of RNA polymerase I [43].

In general, cellular functions of lncRNAs can be divided into (i) directly influencing chromosomes, (ii) modifying chromatin, affecting (iii) transcription, (iv) post-transcriptional modifications, (v) translation and (vi) post-translational modifications [28]. Each of the (i)–(vi) points can be subjected to a separate review, and a detailed description of the cellular functions of lncRNA is not the main subject of this work and can be found elsewhere in many excellent reviews (e.g., [28,29,32,34,44,45,46,47]).

For this review, the cellular functions of lncRNAs can be divided into four categories, in which a lncRNA can act as an antisense, guide, scaffold, or decoy (Figure 3) [48]. Some lncRNAs may perform overlapping functions, but in general, almost all, if not all, known lncRNA cellular functions may obey these modes. In particular, the “Decoy” mode includes recruiting of and interaction with miRNAs (“sponge” or “sink”), but little is known about the interaction between different lncRNAs, which may be important due to their origin and commonness.

## 4. Long Non-Coding RNAs in AMD

The development of bioinformatics resulted in the identification of many lncRNAs that may be important in retinal pathologies, but their functional significance has not been determined in mechanistic studies. In comparison with all identified lncRNAs, very few of them have been attributed specific function(s) in the retina (reviewed in the work of [49,50]).

### 4.1. Long Non-Coding RNAs in Retinal Development

Many transcription factors that are critical for the development of the retina have natural antisense transcript or opposite sense transcript associated with their transcriptional units [51,52]. These include Six3 (SIX homeobox 3) and Six6, Pax6 (paired box 6) and Pax 2, Vax2 (ventral anterior homeobox 2), Otx2 (orthodenticle homeobox 2), and Rax (retina and anterior neural fold homeobox).

The opposite strand transcript Vax2os was shown to play an important role during retinal development with the strongest expression of the Vax2os1 isoform [53]. Vax2os1 expression, like the sense Vax2 transcript, was localized in the ventral retina and downregulated during postnatal stages. However, Vax2os1 is principally expressed in the outer neuroblastic layer and is upregulated in adult animals in the ventral outer nuclear layer.

Six3os is expressed in the retina from prenatal stages to maturity [51,54]. Six3os is essential for the right specification of bipolar, photoreceptor, and Müller glia cells in the developing retina that are regulated by its corresponding sense transcription factor [54].

lncRNA TUG1 (taurine upregulated gene 1) is expressed in the developing and adult retinal tissues to promote the production of rod photoreceptors [55]. Knockdown of the *TUG1* gene in the newborn retina resulted in malformed or absent photoreceptor outer segments. Moreover, the transfected rod photoreceptors exhibited impaired migration into the outer nuclear layer, ectopic expression of cone-specific proteins, and augmented apoptosis. Therefore, TUG1 is necessary for the proper production of photoreceptors in the developing retina. In the adult retina, *TUG1* is mainly expressed in the inner nuclear layer.

The myocardial infarction-associated transcript (MIAT, also known as retinal non-coding RNA2, RNCR2) is an abundant lncRNA with a punctuate distribution in the nucleoplasm [56]. In the neural retina, MIAT is expressed in differentiating progenitor cells with no expression in the adult retina [52]. MIAT knockdown resulted in an increase in the number of amacrine interneurons and Müller glial cells, suggesting that it might be important for type-specification of retinal cells [57].

Metastasis-associated lung adenocarcinoma transcript 1 (MALAT1) is expressed in all retinal layers, and its expression is increased in stress conditions in Müller cells and retinal ganglion cells [58]. Another lncRNA counteracting stress condition in the retina is MEG3 (maternally expressed gene 3) [59].

In summary, several lncRNAs with identified or putative functions may be associated with retinal development and homeostasis, so their impairment may be involved in retinal pathogenesis, including AMD.

### 4.2. Long Non-Coding RNAs in AMD Patients

Xu et al. observed an upregulated expression of the Vax2os1 (ventral anterior homeobox 2 opposite strand isoform 1) and Vax2os2 lncRNAs in patients with CNV [60]. Vax2os is a lncRNA whose gene sense strand is opposite to the sense strand in a “head-to-head” orientation with the gene encoding the homeobox transcription factor Vax2 [53]. Vax2os1 was the first example of lncRNA acting as a regulator in the mammalian retina during development [53]. Fundus photograph and fluorescein angiograph revealed subretinal hemorrhage, neovascular membrane, and branch retinal vein occlusion in the CNV patients. Optical coherence tomography showed a significant increase in central macular thickness, along with a marked elevation of the RPE. The expression of pigment epithelium-derived factor (PEDF, serpinf1), an anti-angiogenic factor, was markedly downregulated in the aqueous humor of CNV patients. On the other hand, the proangiogenic *VEGF* gene was upregulated in these patients. Altogether, the study of Xu et al. provided evidence for the involvement of lncRNAs in the transcriptional regulatory mechanism to modulate the balance between pro- and anti-angiogenic factors in the eye. However, only 10 wet AMD patients and 10 controls with cataracts were enrolled in that study, but much more extensive research was performed on mice (see the next section).

As stated earlier, oxidative stress-induced dedifferentiation of RPE cells mediated by AKT/mTOR signaling was essential in AMD pathogenesis [12]. The miR-184 miRNA was shown to promote RPE cells proliferation by the suppression of the AKT2/mTOR pathway, indicating the usefulness of mTOR as a therapeutic target in AMD also with the use of epigenetic regulators [61].

Chen et al. evaluated the role of lncRNAs in the differentiation of RPE cells with microarrays [62]. They employed human-induced pluripotent stem cell (iPSC)-derived RPE cells and identified 217 lncRNAs differently expressed along with the differentiation. The ZNF503-AS1 (zinc finger protein 503 antisense RNA 2) lncRNA, located in the cytoplasm of RPE cells, was upregulated along with RPE differentiation and downregulated in the RPE-choroid of atrophic AMD patients. In vitro study confirmed that insufficiency of ZNF503-AS1 inhibited differentiation of iPSC-derived RPE cells and promoted their proliferation and migration. Increased expressions of RPE dedifferentiation markers, including microphthalmia-associated transcription factor (MITF), and nanog homeobox (NANOG), were observed in 30-day-old iPSC-RPE cells transfected with *ZNF503-AS1* siRNA. The *ZNF503-AS1* gene is transcribed from the antisense strand of the *ZNF503* (zinc finger protein 503) locus, so the mechanism of its regulatory action may involve antisense-based *ZNF503* downregulation. ZNF503 may bind both DNA and RNA and act as a transcriptional repressor [63]. Furthermore, nuclear factor κB was identified as a potential transcription factor for *ZNF503-AS1.* Taken together, this study identified the ZXNF503-AS1 lncRNA as a potential biomarker and therapeutic target in dry AMD, and its involvement in the dry AMD pathogenesis may include the dedifferentiation of RPE cells.

In their next work, Chen et al. showed that the *LINC00167* lncRNA gene was downregulated in RPE-choroid samples from AMD patients and dysfunctional RPE cells [64]. This gene was steadily upregulated with RPE cells proliferation. In vitro study showed that inhibition of the *LINC00167* expression resulted in RPE cells dedifferentiation associated with overproduction of mitochondrial ROS and impaired phagocytic ability. Furthermore, LINC00167 functioned as a sponge for miR-203a-3p to restore the expression of the suppressor of cytokine signaling 3 (SOCS3), which further inhibited the Janus kinase (JAK)/signal transducer and activator of transcription (STAT) signaling pathway. Altogether, this study showed the protective role of LINC00167 in AMD pathogenesis by the maintenance of RPE cells differentiation through the modulation of the LINC00167/miR-203a-3p/SOCS3 axis.

The studies of Chen. et al. show that the lncRNAs ZNF503-AS1 and LINC00167 can inhibit dedifferentiation of RPE cells and, in this way, prevent AMD and slow down its progression (Figure 4).

Zhu et al. studied the expression profile in early AMD (without GA or CNV) patients to investigate the role of lncRNAs in AMD pathogenesis [65]. The authors identified 266 differentially expressed genes (94 upregulated and 172 downregulated), including 64 genes encoding lncRNAs. A bioinformatic analysis demonstrated that differentially expressed lncRNAs could play a significant role in visual perception, sensory perception of light stimulus, and cognition. Data from the Kyoto Encyclopedia of Genes and Genomes suggested that the two most influenced pathways were those of phototransduction and purine metabolism. Further analysis with the ARPE-19 cell line revealed that RP11-234O6.2 lncRNA was downregulated in the aging RPE cellular model. Forced overexpression of RP11-234O6.2 resulted in increased cell viability and decreased apoptosis of aging RPE cells. Altogether this study suggests that lncRNAs are differentially expressed in early AMD, and the lncRNA RP11-234O6.2 are involved in the regulation of early AMD and have therapeutic potential. A serious limitation of this study was the small number of enrolled subjects (nine patients and seven controls).

### 4.3. Long Non-Coding RNAs in AMD Models

Because experiments on the live human retina are highly limited, there is a need to perform most AMD-related molecular experiments in model systems. There is not a suitable animal model of AMD, as both mice and rats have no anatomical macula, and retinal degeneration in these species is associated with different consequences than those occurring in humans [66]. The use of non-human primates, although well representing human disease in some cases, is associated with serious problems practically precluding them from routine studies [67]. On the other hand, the ARPE-19 cell line, frequently used in AMD research, represents proliferating and senescent cells, while natural RPE cells in the retina are quiescent, and senescence is an important factor in AMD pathogenesis [68]. Induced pluripotent stem cells obtained from AMD patients and differentiated into RPE cells are a better option for ARPE-19, but as aby model, they also have some disadvantages [69].

Xu et al. constructed an animal model for ocular neovascularization for lncRNA microarray analysis and identified 326/51 lncRNAs that were upregulated/downregulated in the vaso-obliteration or neovascularization phase [60]. They observed that the MAPK (mitogen-activated protein kinase) signaling was the most enriched pathway in both phases.

The inflammatory response is an important factor in AMD pathogenesis (reviewed in the work of [7]). Therefore, any modulation of this response may have potential in AMD prevention and therapy. Kutty et al. showed that the proinflammatory cytokine IFN-γ (interferon gamma) induced expression of the lncRNA BANCR (BRAF (B-raf proto-oncogene, serine/threonine kinase)-activated non-protein-coding RNA, LINC00586) in ARPE-19 cells [70]. This expression was suppressed with JAK inhibitor 1, blocking STAT1 phosphorylation. The role BANCR may play in AMD pathogenesis is not known, but it is reported to regulate epithelial-mesenchymal transition (EMT), which is important in AMD pathogenesis [71,72]. The authors observed changes in the expression of several other lncRNAs in response to proinflammatory cytokines. In summary, BANCR upregulation by IFN-γ through the activation of the JAK/STAT1 signaling pathway in the retina may be important in AMD pathogenesis and further AMD research. The lncRNA BANCR was also shown to be upregulated in ARPE-19 cells challenged with a high concentration of glucose, promoting apoptosis [73]. Silencing BANCR by siRNA inhibited apoptosis in ARPE-19 cells. It was also shown in that study that BANCR was upregulated in DR patients.

Pan and Zhao showed that the lncRNA histone deacetylase 4 antisense RNA 1 (HDAC4-AS1) inhibited HDAC4 expression in human ARPE-19 cells exposed to hypoxic stress [74]. HDAC4-AS1 was upregulated in the hypoxic condition and downregulated after reoxygenation. The authors observed that in hypoxic conditions, HDAC4-AS1 bound the HDAC4 promoter and facilitated the recruitment of hypoxia-inducible factor 1 (HIF-1). As hypoxia-inducible transcription factors were associated with CNV and AMD progression, this regulation may be important in AMD pathogenesis [75].

Sun et al. showed that ARPE-19 cells exposed to hydrogen peroxide or tumor necrosis factor alpha (TNF-α) displayed downregulation of the lncRNA MEG3 [76]. Exposure to H_2_O_2_ and TNF-α aimed to create conditions resulting in dysfunctional RPE cells, typical for AMD. The authors also used human iPSCs and iPSC-derived RPE cells and observed upregulation of MEG3 in these cells along with the differentiation of iPSCs. As MEG3 was mainly expressed in the cytoplasm, the authors speculated that it might act as a sponge for miRNAs, including miR-7-5p, which was important for RPE changes and AMD progression through the interaction with PAX6, which is important for the RPE differentiation by mediating pigmentation and hyperproliferation in the transdifferentiated RPE [77]. In line with that study is the work of Zhu et al., who showed that MEG3 was upregulated after prolonged and intense light exposure in the retinas of 2-month-old male C57BL/6 mice and the 661W murine photoreceptor cell line [59]. MEG3 silencing with shRNA protected against light-induced retinal damage in mice and apoptosis in the photoreceptor cell line. The authors observed that MEG3 regulated the function of photoreceptors by acting as a p53 decoy. MEG3 silencing diminished caspase 3/7 activity, upregulated Bcl-2 (B-cell lymphoma 2 apoptosis regulator), an anti-apoptotic protein, and downregulated Bax (BCL2-associated X, apoptosis regulator). Altogether, these studies show an important role of the MEG3 lncRNA in AMD pathogenesis and ways to modulate it to use in AMD therapy.

MEG3 was also reported to be involved in EMT in DR through its DNMT1 (DNA methyltransferase 1)-mediated methylation and modulation of the PI3K (phosphoinositide 3-kinase)/AKT/mTOR signaling pathway [78]. Although these results are not directly related to AMD, they indicate an important role of MEG3 and the AKT/mTOR signaling in retinal pathologies. MEG3 was shown to be involved in microvascular complications in DR in two other studies [79,80]. Another study showed that MEG3 could relieve apoptosis and inflammation induced by glucose at a high concentration in ARPE-19 cells by suppressing the NF-κB signaling pathway and targeting the miR-34a/SIRT1 (sirtuin 1) axis.

Clusterin (apoliprotein J) is a major protein present in drusen observed in the retinas of AMD patients [81]. It was shown that clusterin decreased ROS production in ARPE-19 cells exposed to hydrogen peroxide and defended them from apoptosis [82]. These protective effects were associated with AKT phosphorylation and suppressed by a PI3K/AKT inhibitor, and the authors concluded that clusterin might play a protective role against the consequences of oxidative stress in human RPE cells via the PI3K/AKT pathway. Suuronen et al. showed that clusterin expression on the mRNA and protein levels was epigenetically regulated in ARPE-19 cells, but the exact mechanism of this regulation was not completely clear [2]. Using a bioinformatic approach, Ye et al. showed a differential expression of 386 lncRNAs and a clusterin-associated ceRNA (competitive endogenous RNA) network with 75 lncRNAs and 32 miRNAs in ARPE-19 cells treated with clusterin the work of [83]. Some of these ncRNAs were previously associated with AMD. Therefore, clusterin may induce lncRNAs and ceRNA network, which may play a role in AMD pathogenesis.

It was proposed that a higher level of the intercellular adhesion molecule (ICAM-1) protein in the macular choriocapillaris might increase the susceptibility of the macula to immune cell-mediated damage in AMD [84]. The ICAM-1 (ICR) related lncRNA is transcribed from the antisense strand overlapping the *ICAM1-ICAM4-ICAM5* gene cluster, which stabilizes the ICAM-1 transcript and increases protein expression [85]. Lumsden et al. showed that the human ICR gene promoter contained binding sites for transcription factors involved in a broad range of pathological stimuli, including hypoxia and metabolic and inflammatory proteins [86]. These stimuli are important factors in AMD pathogenesis. Then, the authors verified that human retinal endothelial cells expressed *ICR* and that expression was induced by TNF-α.

Yu et al. treated ARPE-19 cells with multiple stressful factors, including tert-butylhydroperoxide (t-BuOOH), an oxidative stress inducer, to mimic AMD conditions [87]. Multiple stress-induced ARPE-19 cell death and upregulation of the lncRNA PWRN2 (Prader-Willi region non-protein-coding RNA 2) were observed in that study. Using siRNA technology and the pcDNA overexpression system, the authors constructed ARPE-19 sublines to upregulate and downregulate PWRN2. PWRN2 downregulation protected against apoptosis and multiple mitochondrial impairments in t-BuOOH-exposed cells, and PWRN2 upregulation aggravated cellular damage in these cells. That study pointed at the PWRN2 lncRNA as a potential factor in AMD pathogenesis and supported the role of epigenetic regulation in this disease.

MALAT1 is one of the earliest discovered and best-characterized lncRNAs with its primary role in cancer (reviewed in the work of [88]). However, it was also reported to be involved in retinal pathologies. Yang et al. showed that MALAT1 was involved in the transforming growth factor beta (TGF-β)-induced EMT in ARPE-19 cells [89]. MALT1 silencing resulted in the inhibition of TGF-β-induced EMT, migration, and proliferation of RPE cells. Yao et al. showed an upregulation of MALAT1 in the retinas of optic nerve transection (ONT) rat and mouse models [58]. In vitro studies showed that MALAT1 was upregulated in cultured retinal Müller cells and primary retinal ganglion cells (RGCs) under stress conditions induced by hypoxia, high glucose, hydrogen peroxide, and excitatory toxicity of glutamate treatment. The authors showed that MALAT1-CREB (cyclic AMP-responsive element-binding protein 1 binding retains CREB phosphorylation by inhibition of PP2A (protein phosphatase 2)-mediated dephosphorylation, leading to permanent CREB signaling activation. Therefore, the lncRNA MALAT1 may be involved in stress-induced retinal degeneration through the CREB signaling.

miR-125b was reported to inhibit neovascularization through translational suppression of VE (vascular endothelial)-cadherin in endothelial cells [90]. Liu et al. showed an upregulation of MALAT1 and VE-cadherin and downregulation of miR-125b in human retinal microvascular endothelial cells (hRMECs) treated with glucose at high concentration [91]. The authors concluded that MALAT1 could compete with VE-cadherin to bind to miR-125b. Binding of this miRNA in the 3′-untranslated region (3′-UTR) of VE-cadherin resulted in its upregulation. They also observed that knockdown of MALAT1 inhibited proliferation, migration, and angiogenesis of hRMECs cells. Although the authors performed their research to explore mechanisms of DR, their results are important for retinal neovascularization in general and indicate MALAT1 as a potential target in retinal neovascularization-related diseases, including AMD.

XIST is likely the most prominent lncRNA among all known so far (reviewed in the work of [46]). It is responsible for one X chromosome inactivation in females to compensate for the presence of only one X chromosome in males. It was shown downregulated in ARPE-19 cells that were stressed by high glucose concentration and displayed increased apoptosis and decreased migration [92]. XIST overexpression protected ARPE-19 cells against high glucose-induced stress by decreasing apoptosis and restoring migration capability. XIST bound and inhibited miR-21-5p in these cells, suggesting that it might act as a sponge for that miRNA. These interactions may support the suggestion on female sex as a putative ADM risk factor [93].

MIAT (RNCR2) is a lncRNA that aberrant expression was observed in many human diseases, including myocardial infarction, schizophrenia, ischemic stroke, diabetic complications, age-related cataract, and cancer [94]. Jiang et al. showed that MIAT was downregulated in the vaso-obliteration stage and upregulated in the neovascular stage in the retina of the mouse model of oxygen stress-induced retinopathy [95]. These results were confirmed in Müller cells and neurons also in hypoxic conditions and extended through the involvement of MIAT in the MIAT/miR-150-5p/VEGF network in neurovascular dysfunction. Altogether, these results indicate the potential involvement of MIAT in AMD pathogenesis and its therapeutic significance.

To explore the function and therapeutic potential of lncRNAs in CNV, Zhang et al. evaluated the expression profile of mRNAs and lncRNAs in a mouse laser-induced model of CNV by microarray analysis [96]. They identified 716 lncRNAs and 821 mRNAs differently expressed in CNV mice as compared to controls. To identify the lncRNA-mRNA interaction, they constructed a coding-non-coding gene co-expression (CNC) network based on seven validated altered lncRNAs (uc009ewo.1, AK148935, uc029sdr.1, ENSMUST00000132340, AK030988, uc007mds.1, ENSMUST00000180519) and 282 interacted and altered mRNAs, connected by 713 edges. Analysis by Gene Ontology and Kyoto Encyclopedia of Genes and Genomes revealed that altered mRNAs and lncRNAs were enriched in the immune system processes and the chemokine signaling pathways. This study supported the role of lncRNAs-mediated immunological regulation in wet AMD development. In their previous work, Zhang et al. evaluated the lncRNA potential in neovascularization by studying lncRNA and mRNA expression profile in a mouse model of oxygen-induced retinopathy (OIR) by microarrays [97]. They identified 198 upregulated and 175 downregulated lncRNAs in OIR mice compared to controls. They also identified 412 upregulated and 127 downregulated mRNAs. By the further analysis of four validated lncRNAs (ENSMUST00000165968, ENSMUST00000153785, ENSMUST00000134409, and ENSMUST00000154285) and the nearby coding gene, the authors concluded that the interacted mRNAs were mainly enriched in blood vessel development, angiogenesis, cell adhesion molecules, and leukocyte transendothelial migration pathways. The conclusions on the involvement of lncRNAs in the CNV were confirmed in another work with the use of essentially the same attitude and methods [98].

## 5. Conclusions and Perspectives

Long non-coding RNAs are an emerging issue in molecular biology and clinical sciences, reflecting an emerging interest in the epigenetic regulation of gene expression as a main mechanism determining normal and disease phenotypes. Many new lncRNAs are identified with potentially important functions in physiology and pathology.

Recently, an impressive increase in the number of works on the role of epigenetics in AMD pathophysiology is observed. Although DNA methylation and histone modifications dominate in these studies, non-coding RNAs are also subjected. This is in line with the revealing increasing role of the human transcriptome in cellular homeostasis [99]. The collection of lncRNAs, which are important in retinal development and/or AMD pathophysiology, also increases. These lncRNAs may exert protective effects against AMD development or aggravate detrimental changes in the retina contributing to the AMD pathology (Table 1). In either case, they could be potentially targeted in AMD prevention and therapy.

As presented above, in many cases, the mechanism of the involvement of a lncRNA in AMD pathogenesis included its action as a sponge for miRNAs. Therefore, studies on lncRNAs in AMD may bring important information on the involvement of miRNAs in this disease, which is also an emerging issue in research on AMD pathogenesis [22,100].

lncRNAs are a novel class of functional RNA; the landscape of their mutations and variations is small as compared with other ncRNAs, not to mention mRNAs. However, the variability of lncRNA-encoding genes in the pathogenesis of human diseases, especially in cancer, is emerging (reviewed in the work of [101]), but we have not found any association of lncRNA we described in this review with AMD. On the other hand, AMD is reported to associate with mutations in hundreds of genes, often in the form of polymorphisms, which should be considered in experimental studies and projection of therapeutic interventions (reviewed in the work of [102]).

As mentioned, currently, there is not a reliable animal model for AMD, although several well-designed attempts with some successes have been made (reviewed in the work of [103]). Most of the lncRNAs presented in this review were investigated in humans. Therefore, it is important to check the homology between human lncRNAs and their counterparts in AMD animal studies.

lncRNA dynamics is another subject that should be addressed in studies on lncRNA in AMD as lncRNAs undergo post-translational surveillance and only a small subset of the transcripts attain sufficient stability to persist in the cellular environment and perform their functions (reviewed in the work of [104])

As stated earlier, AMD prevalence was expected to progressively increase. However, some studies, including large cohort projects, suggest a decreasing generational trend in the 5-year incidence of an early form of AMD [105,106,107]. This trend is also significant after standardization on age, smoking, sex, and other risk factors. It is presumed that this tendency reflects changes in environment and lifestyle during the last decades [108]. Again, the influence of environmental and lifestyle changes on the epigenome may be more significant than on the genome [19].

Several works on the involvement of lncRNA in AMD pathogenesis are mostly association studies with the identification of a single lncRNA or lncRNA set supported by advanced bioinformatics analysis to identify a network of lncRNA-mRNA interaction to draw a conclusion on the role the identified lncRNAs might play in AMD pathogenesis. No mechanistic and clinical studies are performed to validate these assumptions. Such investigations are conducted in silico, and therefore they predict candidate lncRNAs, some of which may not be confirmed in biological studies. Therefore, mechanistic and clinical studies are needed to validate the biological functions of already identified and new lncRNAs. In addition, there are critical demands for well demographically characterized clinical material to perform a longitude analysis to better understand the role of lncRNA in AMD progression.

## Figures and Tables

**Figure 1 ijms-22-09178-f001:**
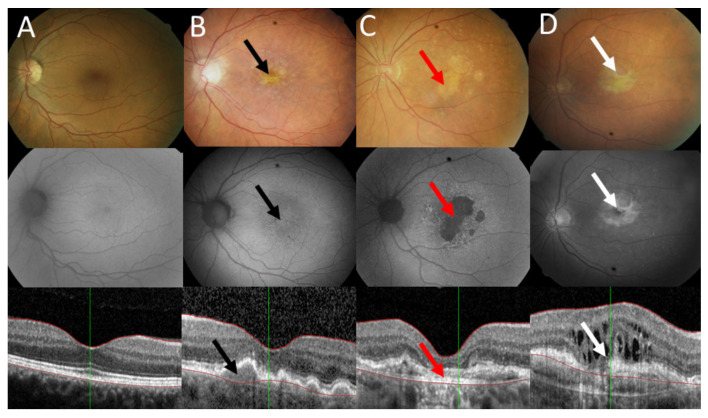
Clinical presentation of age-related macular degeneration. Fundus photograph (upper panel), autofluorescence (middle panel), and optical coherence of normal eye (**A**), intermediate dry AMD with drusen (black arrows) (**B**); advanced geographic atrophy with typical GA lesions (red arrows) (**C**); and wet AMD with subretinal and with fibrotic signs (white arrows) (**D**).

**Figure 2 ijms-22-09178-f002:**
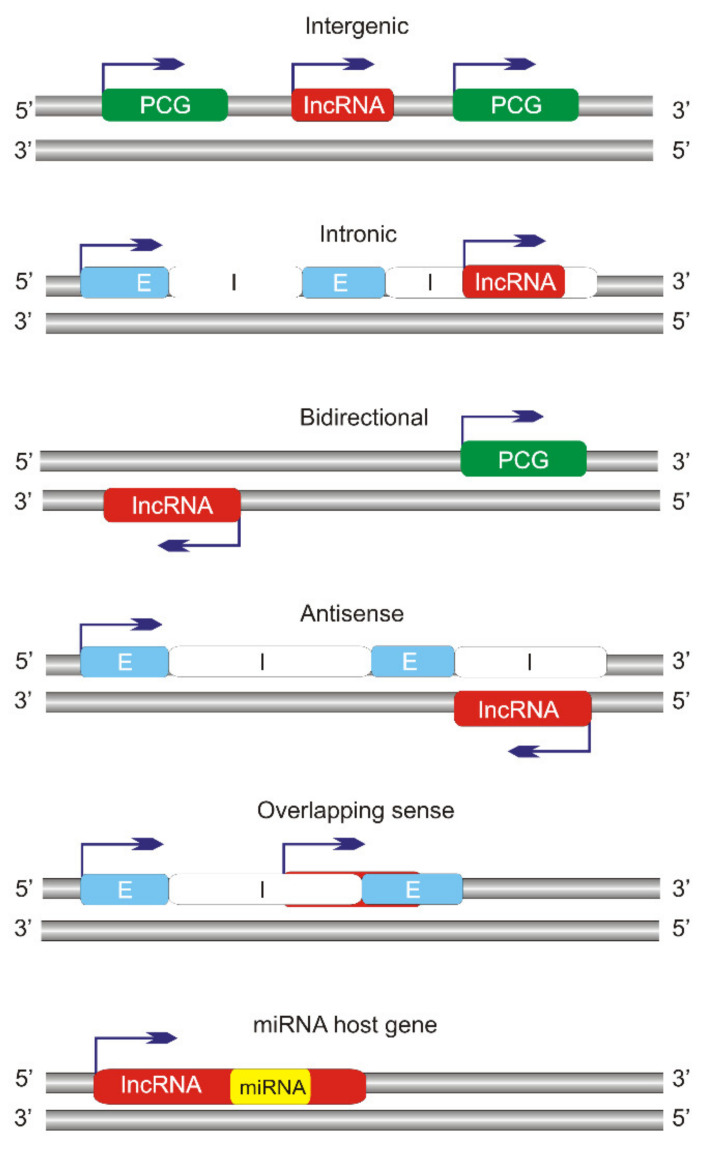
Classification of long non-coding RNAs (lncRNAs) according to orientation and location of their genes with respect to protein-coding genes (PCGs). Introns and exons are designated as I and E, respectively. Intergenic lncRNAs do not overlap, even in part, with any PCG, and they are also called long or large intergenic non-coding (linc) RNAs. Intronic lncRNAs are located within PCG introns and may adopt many shapes, including circular intronic (ci) RNAs produced from lariat introns. Bidirectional lncRNAs originate from the opposite strand to transcribed PGC strand. Antisense lncRNAs are a common feature of the human genome and are “natural” antisense transcripts (NATs). Overlapping sense lncRNAs are located within PCGs with overlapping exons and are transcribed in the same sense direction as PCGs. Micro RNAs (miRNAs) or a cluster of miRNA genes can host a lncRNA gene.

**Figure 3 ijms-22-09178-f003:**
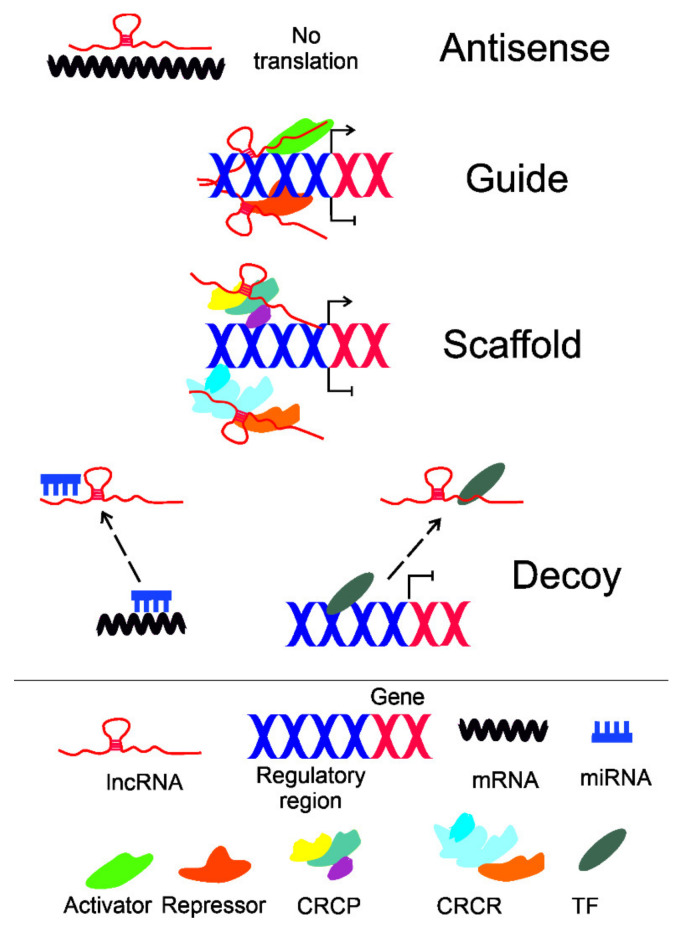
Basic modes of the cellular action of long non-coding RNA (lncRNA) in the regulation of gene expression: antisense, guide, scaffold, and decoy. A lncRNA may pair with a complementary fragment of mRNA, preventing or inhibiting its translation. A lncRNA recruits and/or guides transcriptional activators and repressors to activate/repress transcription of the target gene. A lncRNA may serve as a platform (scaffold) to facilitate assembling a chromatin remodeling complex to change the structure of chromatin into a more open (CRCP—chromatin remodeling complex acting permissively) or closed (CRCR—chromatin remodeling complex acting repressively) configuration. A lncRNA may act as a decoy to recruit (broken arrows) micro RNAs (miRNAs) or transcriptional factors (TFs) and sequester them from their target mRNA or DNA, respectively. Presented are only examples of lncRNAs actions in gene expression regulation, and many other mechanisms, e.g., those related to translation and post-translational regulations, are not illustrated, but in general, they follow the presented schemes. Some modes of action of lncRNAs may overlap.

**Figure 4 ijms-22-09178-f004:**
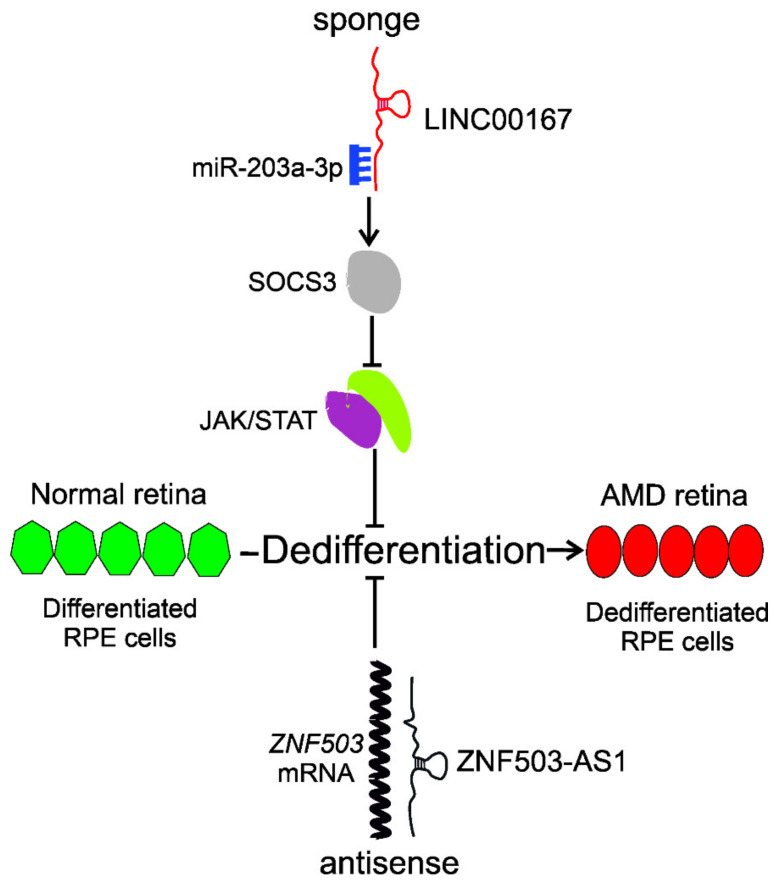
Long non-coding RNAs (lncRNAs) ZNF503-AS1 (zinc finger protein 503 antisense RNA 2) and LINC00167 inhibit the dedifferentiation of retinal pigment epithelium (RPE) cells, important in age-related macular degeneration (AMD). These lncRNAs act as an antisense inhibitor and sponge for micro RNAs, respectively. ZNF503-AS1 targets mRNA of the *ZNF503* gene as it has a complementary sequence and prevents its translation resulting in the inhibition of dedifferentiation of RPE cells. LINC00167 acts as a sponge for miR-203a-3p restoring the activity of the suppressor of cytokine signaling 3 (SOCS3), which further inhibited the Janus kinase (JAK)/signal transducer and activator of transcription (STAT), resulting in the inhibition of dedifferentiation of RPE cells.

**Table 1 ijms-22-09178-t001:** Effects and mechanisms of long non-coding RNAs (lncRNAs) in retinal cells potentially important in age-related macular degeneration (AMD).

lncRNA	Effect	Mechanism	Reference
Vax2os1/Vax2os2 *^a^*	CNV inhibition	PEDF upregulation, VEGF downregulation	[60]
ZNF503-AS1	Inhibition of dedifferentiation of RPE cells	Antisense for ZNF503	[62]
LINC00167	Inhibition of dedifferentiation of RPE cells	Sponge for miR-203a-3p, restore SOCS3, inhibiting JAK/STAT signaling	[64]
BANCR	RPE protection	Upregulation by IFN-γ, interaction with JAK/STAT1, EMT inhibition in RPE cells	[71,72]
HDAC4-AS1	CNV induction	Binding of the *HDAC4* promoter and recruitment HIF-1	[75]
MEG3	RPE protection	Sponge for miRNAs, decoy for p53, downregulating Bcl-2 and upregulating Bax, targeting the NF-κB-miR-34a-SIRT1 axis; inhibition of EMT through its DNMT-mediated methylation and modulation of the PI3K/AKT/mTOR signaling	[79,80]
ICR	Protection of the retina	Decoy for many transcription factors involved in pathological stimuli	[86]
PWRN2	Aggravating RPE damage in stress conditions	Largely unknown, putative prooxidant effects	[87]
MALAT1	Aggravating retinal cell damage in stress conditions; CNV induction	Stimulation of TGF-β-induced EMT; modulation of the PP2A-CREB axis; sponge for miR-125b	[58,89,91]
XIST	RPE protection	Antioxidant effects, sponge for miR-21-5p	[92]
MIAT	Retinal neovascularization	Involvement in miR-150-5p/VEGF network to promote neurovascular dysfunction	[95]

*^a^*—all abbreviations are defined in the main text.

## Data Availability

The data that support the findings of this study are available from the corresponding author upon reasonable request.

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
