# Peer review of "Potential of Long Non-Coding RNAs in Age-Related Macular Degeneration"

_ijms, 2021, doi:10.3390/ijms22179178_

Round 1

Reviewer 1 Report

This manuscript by Blasiak et al. reviewed the potential of long non-coding RNAs (lncRNAs) in AMD. It is suitable for publication in IJNS with minor revision. I only have some questions that need to be addressed before publication.

 .

  1. The authors showed that some lncRNAs maybe developmentally regulated. If some lncRNAs in this manuscript also show that there is some gender, racial or age-related differences in the literature, please briefly discuss it.
  2. If these lncRNAs in this manuscript also show that there is some mutations, or single nucleotide polymorphisms (SNPs), or copy number variation (CNV) in these lncRNAs, please give brief discussion, because some AMD patients are associated with gene mutations, some gene mutations may be located at the gene promoter regions, which are the targets of some lncRNAs.
  3. Are lncRNAs discussed in this manuscript highly conserved among mouse, rat, NHP, or human? If so, please give a brief discussion.
  4. Do these lncRNAs have a short turnover or half life? Or are they vey stable? Please give a brief discussion regarding its quick turnover and functional regulation of mRNAs in its host genes.
  5. From line 446, Some of these ncRNA were previously associated with AMD. Is it “ncRNA” or “lncRNA”?
  6. From line 211, further studies revealed that that lncRNA regulated CDKN1A in cis. Is “that that” ok?
  7. From line 534, Long non-coding RNAs is an emerging issue in molecular biology and clinical sci-.

You may change “RNAs is” to “RNA is” or to “RNAs are”?

  1. From line 278, The myocardial infarction associated transcript (MIAT, retinal non-coding RNA2.

Please change “MIAT, retinal non-coding RNA2” to “MIAT, also known as retinal non-coding RNA2”.

Thank you for the invitation!

Author Response

This manuscript by Blasiak et al. reviewed the potential of long non-coding RNAs (lncRNAs) in AMD. It is suitable for publication in IJNS with minor revision. I only have some questions that need to be addressed before publication.

Comment: The authors showed that some lncRNAs maybe developmentally regulated. If some lncRNAs in this manuscript also show that there is some gender, racial or age-related differences in the literature, please briefly discuss it.

Answer: We are sorry, but we do not have information on the gender- or racial-dependent expression of lncRNAs that are listed in Table 1. These lncRNAs whose expression is important in the retina development are described in the main text.

Comment: If these lncRNAs in this manuscript also show that there is some mutations, or single nucleotide polymorphisms (SNPs), or copy number variation (CNV) in these lncRNAs, please give brief discussion, because some AMD patients are associated with gene mutations, some gene mutations may be located at the gene promoter regions, which are the targets of some lncRNAs.

Answer: We have added the following fragment to the concluding section:

“lncRNAs are a relatively novel class of functional RNA, the landscape of their mutations and variations is small as compared with other ncRNAs, not to mention mRNAs. However, the variability of lncRNA-encoding genes in the pathogenesis of human diseases, especially in cancer, is emerging (reviewed in [102]), but we have not found any association of lncRNA we described in this review with AMD. On the other hand, AMD is reported to associate with mutations in hundreds of genes, often in the form of a polymorphisms, which should be taken into account in experimental studies and projection of therapeutic interventions (reviewed in [103]).”

with new references:

  1. Aznaourova, M.; Schmerer, N.; Schmeck, B.; Schulte, L.N. Disease-Causing Mutations and Rearrangements in Long Non-coding RNA Gene Loci. Frontiers in genetics 2020, 11, 527484, doi:10.3389/fgene.2020.527484.
  2. de Jong, S.; Gagliardi, G.; Garanto, A.; de Breuk, A.; Lechanteur, Y.T.E.; Katti, S.; van den Heuvel, L.P.; Volokhina, E.B.; den Hollander, A.I. Implications of genetic variation in the complement system in age-related macular degeneration. Progress in retinal and eye research 2021, 100952, doi:10.1016/j.preteyeres.2021.100952.

Comment: Are lncRNAs discussed in this manuscript highly conserved among mouse, rat, NHP, or human? If so, please give a brief discussion.

Answer: We have added the following fragment to the concluding section.

“As mentioned, currently there is not a reliable animal model for AMD, although several well-designed attempts with some successes have been made (reviewed in [104]). Most of lncRNAs presented in this review were investigated in humans. Therefore, it is important to check the homology between human lncRNAs and their counterparts in AMD animal studies.”

with new reference:

  1. Ratnayaka, J.A.; Lotery, A.J. Challenges in studying geographic atrophy (GA) age-related macular degeneration: the potential of a new mouse model with GA-like features. Neural Regen Res 2020, 15, 863-864, doi:10.4103/1673-5374.268972.

Comment: Do these lncRNAs have a short turnover or half life? Or are they vey stable? Please give a brief discussion regarding its quick turnover and functional regulation of mRNAs in its host genes.

Answer: We have added the following sentence to the concluding section:

“lncRNA dynamics is another subject that should be addressed in studies on lncRNA in AMD as lncRNAs undergo post-translational surveillance and only a small subset of the transcripts attain sufficient stability to persist in the cellular environment and perform their functions (reviewed in 105)”

with new reference

  1. Nair, L.; Chung, H.; Basu, U. Regulation of long non-coding RNAs and genome dynamics by the RNA surveillance machinery. Nature reviews. Molecular cell biology 2020, 21, 123-136, Comment: From line 446, Some of these ncRNA were previously associated with AMD. Is it “ncRNA” or “lncRNA”?

Comments: From line 211, further studies revealed that that lncRNA regulated CDKN1A in cis. Is “that that” ok?

From line 534, Long non-coding RNAs is an emerging issue in molecular biology and clinical sci-.

You may change “RNAs is” to “RNA is” or to “RNAs are”?

From line 278, The myocardial infarction associated transcript (MIAT, retinal non-coding RNA2.

Please change “MIAT, retinal non-coding RNA2” to “MIAT, also known as retinal non-coding RNA2”.

Answer: All these remarks have been addressed.

Reviewer 2 Report

AMD is the leading cause of visual impairment in elderly people. A huge number of genetic studies have been carried out to try and discover risk factors and possible target for genetic therapy, but unfortunately much less has revealed effective in real life.

This manuscript is an interesting review about studies on genetics and epigenetics over last decades. General organization and references are well-done. Nothing is really redundant. Figures and tables are a valuable attempt to summarize this complicated issue. Focusing on long non-coding RNAs, as stated in Conclusion, might be an interesting issue where molecular biology and genetics could be support clinical medicine. However, we are too far away from effective results and a lot of studies still have to be carried out.

Author Response

AMD is the leading cause of visual impairment in elderly people. A huge number of genetic studies have been carried out to try and discover risk factors and possible target for genetic therapy, but unfortunately much less has revealed effective in real life.

Comment:This manuscript is an interesting review about studies on genetics and epigenetics over last decades. General organization and references are well-done. Nothing is really redundant. Figures and tables are a valuable attempt to summarize this complicated issue. Focusing on long non-coding RNAs, as stated in Conclusion, might be an interesting issue where molecular biology and genetics could be support clinical medicine. However, we are too far away from effective results and a lot of studies still have to be carried out.

Answer: Thank you.